# A Review on Biomedical, Biomolecular, and Environmental Monitoring Applications of Cysteamine Functionalized Nanomaterials

**DOI:** 10.3390/mi16101144

**Published:** 2025-10-08

**Authors:** Muthaiah Shellaiah

**Affiliations:** National Research Institute of Chinese Medicine, Ministry of Health and Welfare, Taipei 11221, Taiwan; muthaiah1981@nricm.edu.tw

**Keywords:** cysteamine, free thiol and amine, nanomaterials, colorimetric detection, luminescence sensor, environmental monitoring, biomedical scrutiny, biomolecular detection, real-time applications, hazard quantification

## Abstract

Functionalizing agents enhance the photophysical properties of nanomaterials, thereby broadening their applications. Among these agents, cysteamine (SH-(CH_2_)_2_-NH_2_) is unique because of its free thiol (-SH) and amino (-NH_2_) groups. The presence of free -SH or -NH_2_ groups significantly enhances the functionalization of highly stable nanomaterials. These stable nanomaterials, which contain free -SH or -NH_2_ groups, can effectively bind with biomedical, biomolecular, and environmental analytes, improving sensor performance and making them valuable materials. In this context, cysteamine-functionalized nanoparticles (NPs), quantum dots (QDs), nanoclusters (NCs), nanocomposites, and other nanostructures have been demonstrated to be useful for quantifying biomedical, biomolecular, and environmental analytes. To date, no review has outlined the functionalizing ability of cysteamine or the application of cysteamine-functionalized nanomaterials in biomedical, biomolecular, and environmental analyte monitoring. This review emphasizes the role of cysteamine in producing stable nanomaterials and detecting specific biomedical, biomolecular, and ecological analytes. It also covers general protocols for functionalizing with cysteamine, the mechanistic basis of analyte detection, and their advantages, limitations, and prospects.

## 1. Introduction

Monitoring biological and environmental species with nanomaterials has become an area of interest [1,2,3]. Numerous research units have reported using nanomaterials to detect biomarkers, biomolecules, drugs, nucleic acids (DNA and RNA), and specific diseases [4,5,6,7,8,9]. The effective quantification of environmental hazards, such as heavy metal ions, toxic anions, pesticides, herbicides, volatile organic compounds (VOCs), and nitroaromatic compounds (NACs), is facilitated by nanomaterials [10,11,12,13,14,15]. However, their sensitivity and selectivity depend on their structural stability and photophysical and opto-electronic properties [16,17,18]. Using the functionalizing/capping agents could improve the stability of nanomaterials and their photophysical and opto-electronic properties [19,20,21]. For example, the greater affinity of mercaptans and free thiol groups to noble metals such as gold (Au), silver (Ag), and Copper (Cu) leads to the formation of stable nanoparticles, nanoclusters, nanorods, nanoprism, etc., with improved colorimetric or fluorometric properties [22,23,24].

The surface charge potential influences the interaction of analytes with nanomaterials, thereby affording sensor responses [25,26,27]. While employing the functionalizing/capping agents with distinct functional units such as thiols (-SH), amine (-NH_2_ and -NH), carboxyl (-COOH), and amides/thioamides (-CO-NH and -CS-NH), the surface charge potential of nanomaterials can be fine-tuned to improve the analyte interaction [28,29,30,31,32]. Thus, it has been stated that the selection of a functionalizing agent may determine the effectiveness of nanomaterials in the sensor studies [33,34]. In this way, employing the functionalizing moiety with a dual functional unit may be crucial in detecting a specific analyte. For instance, Shellaiah et al. used glutamic acid (which holds a free amine (-NH_2_) and two carboxyl (-COOH) functional units) as a stabilizing agent to synthesize stable gold nanoparticles (Glu-AuNPs) for detecting Cr^3+^ ions at a sub-nanomole detection limit (LOD) at pH 6 [35]. The free -NH_2_ becomes -NH_3_^+^ at pH 6 and attaches over the -Vely-charged bare AuNP surface, leaving the two -COO- groups freely to coordinate with Cr^3+^ ions. Likewise, biothiol (cysteine/homocysteine/Glutathione)-functionalized nanomaterials display effective sensor response to diverse analytes [36,37,38,39,40,41,42]. The free -SH affords strong binding to the nanomaterial’s surface and avails the -COOH and -NH_2_ groups for analyte interaction. Interestingly, the pH values can tune the surface charge potential of mercaptans and biothiol-functionalized nanomaterials [43,44]. 

The use of cysteamine (CA, containing free -NH_2_ and -SH groups connected via a -CH_2_-CH_2_ spacer) for functionalizing nanomaterials in biomedical, biomolecular, and environmental monitoring has been reported in the literature [45]. When the cysteamine’s free -SH group is attached to the nanomaterial surface, the -NH_2_ can interact with analytes to produce a sensor response. Conversely, while the -NH_2_ is functionalized on the nanomaterial, the -SH group remains available for analyte detection. The functionalization of the -SH and -NH_2_ groups can vary depending on pH levels [45]. In fact, the surface charge of -SH (of CA)-functionalized nanomaterials may be positive (+Ve) due to the protonation of -NH_2_ to form -NH_3_^+^ [45,46,47]. Conversely, the charge of -NH_2_ (of CA)-functionalized nanomaterials may be either positively or negatively charged depending on the pH conditions [48,49,50]. Many studies utilize cysteamine-functionalized nanomaterials, such as nanoparticles, nanoclusters (NCs), quantum dots (QDs), and nanocomposites, for detecting and quantifying biomedical, biomolecular, and environmental contaminants. These detections were carried out using various methods, such as colorimetric, fluorometric, electrochemical, and electrochemiluminescence techniques. This review provides comprehensive information on the applications of cysteamine-functionalized nanomaterials in biomedical, biomolecular, and environmental monitoring, as shown in Figure 1. 

## 2. Biomedical and Biomolecular Monitoring Applications

Jamaluddin et al. used the cationic cysteamine-functionalized gold nanoparticles (CA@AuNPs) for optical reflectometry-based detection of SARS-CoV-2 (COVID-19) [51]. The cationic CA-Au NPs exhibit an optical reflectance maximum at 639 nm, resulting in a red-colored solution. When interacting with DNA, the CA@AuNPs undergo aggregation, changing color from red to purple. This occurs due to the strong affinity between the negatively charged sugar-phosphate groups in DNA and the positively charged CA@AuNPs. During this process, the optical reflectance peak shifts from 639 nm to 765 nm. The addition of SARS-CoV-2 RNA to the system causes anti-aggregation, resulting in a reversal of the color and optical reflectance. The COVID-19 RNA shows a linear response between 25 nM and 200 nM (nM = nanomole; 10^−9^ M) with a limit of detection (LOD) of 0.12 nM. Results from this method become comparable to operable PCR (Polymerase Chain Reaction) COVID-19 detection. This approach can also identify asymptomatic cases and enable early diagnosis of COVID-19. However, to improve the detection limit, extensive research becomes mandatory. Importantly, it is crucial to address the following question: ‘How could this method become feasible in commercial devices?’

Farshchi et al. combined iron oxide nanoparticles (Fe_3_O_4_NPs; size = 62–81 nm) and CA@AuNPs to enable electroconductive interface-based discrimination of prostate-specific antigen (PSA) for early cancer diagnosis [52]. The SWV response of PSA becomes linear from 0.001 to 1 μg/L (µg = microgram; 10^−6^ g) with a LOD of 0.001 μg/L. Although the results are promising, they lack mechanistic insights, details about the role of CA, and application in real-time cancer diagnosis. Therefore, the commercialization and practical use of this method remain uncertain. Cysteamine was functionalized onto AuNPs via a strong Au-S bond, leaving the free -NH_2_ group to react with folic acid (FA), producing FA-CA@AuNPs (size = 39 nm; zeta potential (ζ) = −35.1 mV) for cytosensing, monitored through differential pulse voltammetry (DPV) signals [53]. This approach demonstrated the recognition of HT29 cancer cells in HEK293 FR-negative cells (FR = folate receptor). From 250 to 5000 cells/mL (mL = milli liter; 10^−3^ L), the HT 29 cells were detected with a lower quantification limit (LLOQ) of 250 cells/mL. Cysteamine is crucial in conjugating AuNPs and FRs at both ends. The study showcases cellular imaging, pH conditions (pH 7.4), cytometry, and reproducibility assessments. This technique could be applied to developing portable biomedical devices. However, the reliability of such devices depends on the stability of CA@AuNPs in real-world environments. 

Hasanzadeh et al. fabricated a glassy carbon electrode (GCE) with porous cysteamine-functionalized graphene quantum dots (GQDs) through the NH_2_ group of cysteamine, leaving the -SH group free, which conjugates with layer-by-layer self-assembled silver nano-cubics (AgNCs) for the quantification of cancer biomarkers [54]. The active surface area of AgNCs has been improved to 8.48 × 10^−6^ cm^2^ s^−1^ due to the binding of CA with the Ag surface; hence, the electron transfer improved during antigen–antibody interaction. The SWV response to cancer antigen 15-3 (CA 15-3) in adenocarcinoma cell lysates shows two linear responses between 0.095–0.5 and 0.25–1 U/mL, with a LOD of 0.019 U/mL. Real sample applications and the stability of the proposed tactic led to extensive utility in biomedical diagnosis. However, this tactic is limited to solution-based analysis and cannot be prescribed for in vivo cellular analysis. Liu et al. proposed a resonance energy transfer (RET)-based strategy for photoelectrochemical cytosensing analysis [55]. RET occurs with carbon dots (CDs), and CA@AuNPs become crucial for photoelectrocurrent (PEC) generation in electrochemical impedance spectroscopy (EIS). Over an indium tin oxide (ITO) electrode, C-dots-CA@AuNPs were cast, followed by the coating of chitosan (CS), immobilization of FA, and bovine serum albumin (BSA) to afford ITO/C-dots–CA@AuNPs/FA/CS/BSA. During cytosensing in HeLa cells, the PEC response in EIS declined from 686 nA to 275 nA (nA = nanoampere; 10^−9^ A). In fact, the interaction of ascorbic acid (AA) present in chitosan was blocked by HeLa cells, resulting in a decline in photocurrent response. The linear PEC response was found from 1 × 10^2^ to 5 × 10^6^ cells/mL with a 15 cells/mL LOD. This work looks impressive in terms of results. However, complications in fabrication and a lack of real-time analysis still cast doubt on its reliability.

Rodríguez et al. pronounced EIS (Nyquist plot)-based sensing of the S100B biomarker associated with traumatic brain injury (TBI) [56]. The lithographic technique deposits Au electrodes (AuEs), followed by a self-assembled monolayer (SAM) of cysteamine, which undergoes cross-link reactions with Anti-S100B (monoclonal antibody) using 1-ethyl-3-(3-dimethylaminopropyl) carbodiimide (EDC), N-hydroxysuccinimide (NHS). A similar procedure was employed to fabricate gold-interdigitated electrodes (AuIDEs). AuEs and AuIDEs show a linear EIS response to S100B in human plasma from 10 to 316 pg/mL (pg = picogram; 10^−12^ g), with LODs of 18 pg/mL and 6 pg/mL. The attained signals correspond to Au/CA/anti-S100B/BSA. This tactic has been stated as a simple, cost-effective protocol that could be extended for biomedical device commercialization. However, using the lithographic tactic to fabricate electrodes in developing or poor economic countries raises concerns about this approach. Electrochemical detection of specific microRNA (miRNA; miR-25) associated with lung cancer was delineated using CA@AuNPs [57]. This nanogenosensor was fabricated by immersing the GCE in CA@AuNPs and glutaraldehyde (GA) to achieve a GA/CA@AuNPs/GCE, which interacts with a single-stranded probe to attain an ss-probe/GA/CA@AuNPs/GCE surface. During the recognition of miR-25, the ss-probe was hybridized to afford an EIS-based sensor response. The Nyquist plot showed two linear responses for miR-25 between the concentration ranges of 1.0 × 10^−12^ to 1.0 × 10^−10^ M and 1.0 × 10^−10^ to 1.0 × 10^−6^ M, with a LOD of 2.5 × 10^−13^ M. The plasma sample analysis showed >97% recovery with <8% RSD values. This approach is impressive but requires sophisticated electrochemical instrumentation and careful optimization. Thus, the reliability in diverse environmental conditions is still in question. 

A pre-activated nylon paper membrane and CA@AuNPs were employed for the peroxidase-mimicking colorimetric detection of MicroRNA-21 (miRNA-21, associated with cancer disease) [58]. The nylon paper was punched into a 6 mm disk, treated with a DNA probe, 3% GA, and 3% BSA. Then, it was hybridized with various concentrations of miRNA-21, washed with buffer, and dried. The +Vely-charged CA@AuNPs were poured and washed with buffer onto the hybridized paper strip. Finally, adding H_2_O_2_ (hydrogen peroxide)/3,3,5,5-tetramethylbenzidine (TMB) solution visualizes the emergence of blue color. The linear colorimetric response to miRNA-21 was perceived between 1 pM (picomole; 10^−12^ M) and 1 nM with a 0.5 pM LOD. This tactic showed 90–97.6% recovery in human serum samples. Too many fabrication steps restrict the use of this tactic. However, this work can be extended to biomedical testing kits with respect to the naked-eye colorimetric response and cost-effective nylon paper strips. A cysteamine, AuNPs, and polyglutamic acid (PGA)-based electrode was fabricated to detect DNA via DPV signals [59]. Immobilization of AuNPs with PGA, cysteamine, and ssDNA via a 5′-thiol-linker resulted in an electrode structure of ssDNA/AuNPs/CA/PGA. The interaction of the target DNA underwent hybridization to afford a DPV response. The linear range for the target DNA was 9.0 × 10^−11^ to 4.8 × 10^−9^ M with a LOD of 4.2 × 10^−11^ M. Here, CA became essential to link two essential electrochemical sensing units. This is an initiative work, reported a decade ago, and lacks valid real-time application. Thus, extensive investigation is required for justification.

Sharma et al. proposed a colorimetric sensing strategy to quantify creatine kinase (CK-MM, linked to cardiac diseases) in the presence of adenosine triphosphate (ATP) via anti-aggregation of CA@AuNPs [60]. In the presence of ATP, the CA@AuNPs underwent a color change from red to blue with a surface plasmon resonance (SPR) peak shift from 532 to 584 nm. The CK-MM inhibited aggregation and restored the color, as shown in Figure 2. The linear range of CK-MM lies between 5.617 × 10^3^ ng/mL and 0.5617 ng/mL (ng = nanogram; 10^−9^ g), with a LOD of 0.569 ng/mL. Due to the CA over AuNPs, the +Ve charge was generated, which plays a vital role in this sensor. This tactic becomes effective in human serum with a LOD of 0.553 ng/mL. While this approach is promising, it falls short in biomedical engineering applications. Therefore, further investigation is necessary for validation.

Jia et al. defined the use of dual-stabilized gold nanoclusters (AuNCs) for detecting carcinoembryonic antigen (Ab_2_) through electrochemiluminescence response [61]. N-acetyl-L-cysteine and cysteamine were employed as capping agents to afford NAC/CA@AuNCs (size = 3 nm), which were conjugated with the target antigen Ab_2_ using EDC and NHS to fabricate an Ab_2_-NAC/CA@AuNCs electrochemiluminescence (ECL) tag. Over the GCE, the signal antibody (Ab_1_) was attached through a P-aminobenzoic acid (ABA) link to achieve a near-infrared (NIR) ECL immunosensor. At pH 7.4, the sandwiched immunocomplex of Ab_2_ generates an NIR ECL response at 860 nm (scan range: 0 to +1.6 V). The linear range lies between 1 fg/mL and 0.5 ng/mL (fg = femtogram; 10^−15^ g) with a LOD of 0.33 fg/mL. This work has been demonstrated in serum samples. However, the role of cysteamine is not clarified in detail. Based on critical evaluation, it is suggested that this tactic requires protracted scrutiny for commercialization. A similar sandwich immunosensor was fabricated by using Fe_3_O_4_NPs and a poly(3,4-ethylenedioxythiophene):poly(styrene sulfonate) conductive composite (Fe_3_O_4_/PEDOT:PSS) for the detection of prostate-specific antigen (PSA, linked to prostate cancer) [62]. The biotinylated antibody (Ab_1_) and the secondary antibody ‘HPR-Ab_2_’ were attached to Fe_3_O_4_/PEDOT:PSS and CA@AuNPs to attain the sandwich immunosensor structure “Fe_3_O_4_/PEDOT:PSS/Ab_1_/BSA/PSA/(CA@AuNPs-Ab_2_)”. The SWV-based immunosensor for PSA showed a linear response from 0.001 to 500 µg/L, with an LLOQ of 0.001 µg/L. Due to the complex fabrication procedures and the lack of real-time applications, the biomedical applications are in question.

Wai et al. proposed an anti-aggregation strategy for colorimetric lysozyme (an anti-microbial agent) recognition using positively charged CA@AuNPs (size = 28 nm and ζ = +44.3 mV) [63]. The CA@AuNPs underwent aggregation while adding negatively charged DNA with a lysozyme-binding aptamer (LBA), changing red to a bluish gray color. In the presence of lysozyme, the LBA present on the surface forms lysozyme/LBA complexed to restore the red color. The best response was observed at 25 nM of LBA. The lysozyme had a linear response range from 37.5 to 180 nM, with an estimated LOD of 2.26 nM. This work is simple without many complications and has high selectivity over other analytes. However, established biomedical device fabrication lacks commercialization. Li and co-workers conjugated the positively charged cysteamine-capped cadmium telluride quantum dots (CA@CdTeQDs: size = 3.1 nm and ζ = +29.5 mV) with negatively charged lysozyme-bound DNA (LBD) via electrostatic interaction, applied in the detection of lysozymes [64]. Upon conjugation of QDs with LBD, the surface charge potential varied from +29.5 mV to −19.7 mV. Ultimately, the interaction of lysozymes with QDs-LBD altered the surface charge potential to −11.26 mV, which enhanced the fluorescence to 592 nm at pH 7.4. The linear detection of lysozymes ranged from 8.9 to 71.2 nM, with an appraised LOD of 4.3 nM. Mouse serum and human urine interrogations showed >90% recovery with <6.5% RSD. This innovation opened a window for biomedical device fabrication. But an extensive investigation is required for defense in terms of photoluminescence quantum yield (PLQY; Φ_F_) and interference analysis.

Ren et al. described the anti-aggregation of CA@AuNPs (size = 34 nm) for detecting glutathione S-transferase (GST), which is associated with oxidative stress [65]. In the presence of GST, glutathione catalyzed the 1-chloro-2,4-dinitrobenzene (CDNB) to afford the CDNB-SG conjugate. In the absence of GST, the free thiol of glutathione adsorbed onto the Au surface and induced aggregation to change color from red to blue. However, in the presence of GST, the color reversed to red via anti-aggregation and CDNB-SG conjugate formation. The linear range of GST was observed between 0.1 and 0.4 U/mL, with an estimated LOD of 0.06 U/mL. This work does not have a valid real-time application. Thus, it can be accounted as a preliminary study. Mradula et al. attached the T4 antibody to CA@AuNPs to detect thyroxine hormone (linked to thyroid disease) [66]. The SPR peak of the anti-T4/CA@AuNPs bioconjugate at 528 nm was linearly quenched by the thyroxine hormone from 0.8 to 100 pM, with a LOD of 11.6 pM. Further investigations are required for mechanistic validation and biomedical applications.

Vasudevan et al. detected the N-acyl homoserine lactone (AHL, linked to urinary tract infections) using cysteamine-capped zinc oxide and titanium oxide nanoparticles (CA@ZnONPs and CA@TiO_2_NPs) [67,68]. The proposed bioassays showed >95% sensitivity to AHL molecules (C4-HSL and 3-oxo-C12 HSL). In artificial urine (AUM), the linear ranges of CA@ZnONPs and CA@TiO_2_NPs to AHL were established as 10–120 nM and 10–160 nM. Both reports were validated by real analysis in a human pathogen, namely ‘Pseudomonas aeruginosa’. Cysteamine plays a linker role and participates in detecting AHL. These reports are impressive but fall short in biomedical engineering. Thus, an extensive investigation is compulsory for biomedical device fabrication. Li and co-workers detected 17β-estradiol (a natural estrogen) using thioglycolic acid-capped CdTe quantum dots (TGA-CdTe QDs) and a CA@AuNPs composite via aptamer-facilitated fluorescence resonance energy transfer (FRET) [69]. The DNA aptamers exhibited affinity for 17β-estradiol, resulting in detailed fluorescent and colorimetric sensor responses. The positively charged CA@AuNPs (size = 40.05 ± 0.58 nm and ζ = +47.2 mV) and negatively charged TGA-CdTeQDs (size = 3.11 nm and ζ = −25.9 mV) formed a FRET conjugate via electrostatic interaction. In the presence of estradiol aptamers, the zeta potential of CA@AuNPs was adjusted to −3.15 mV, with aggregation and a color change. While exposed to 17β-estradiol, the zeta potential value improved to +22 mV, with a dispersion of particles and restoration of color. Further addition of TGA-CdTe QDs to the above system produced colorimetric and fluorometric responses via FRET. The colorimetric linear response of aptamer Au-QDs at 527 nm was perceived from 1 ng/mL to 85 ng/mL, with a LOD of 0.77 ng/mL. Similarly, the FRET-based linear emission quenching at 538 nm was found between 0.5 ng/mL and 150 ng/mL, with a LOD of 0.057 ng/mL. Spiked real analysis validated this tactic by comparing it to HPLC studies. With respect to its negligible interference, real-time applications, and unique mechanism, this work can be stated as innovative. Still, biomedical engineering requires more information on the fluorescent stability of composites in diverse environmental conditions.

A cysteamine-functionalized nanogold (CA-Au; size = 65 nm) and 4-mercaptobenzoic acid-modified gold (4-MBA-Au; size = 55 nm) electrode were fabricated for the electrochemical detection of Brucella (associated with bacterial infection) in milk samples [70]. The SWV response to Brucella had a linear regression between 1.6 × 10^2^ and 1.6 × 10^8^ cfu/mL (cfu = colony-forming units), with a LOD of 5.12 × 10^2^ cfu/mL. The CA-Au was cross-linked over a 4-MBA self-assembled screen-printed electrode (SPE) and played a crucial role in sensing Brucella antibodies. This biosensor showed high reproducibility and stability. The analysis of spiked milk samples demonstrated >95% recovery, validating this approach. However, biomedical devices based on this method require careful optimization. To detect 8-Oxo-7,8-dihydro-2′-deoxyguanosine (8-oxo-dG; a major product formed during reactive oxygen species (ROS)-induced DNA damage), anti-aggregation of CA@AuNPs (size = 33.84 ± 3.47 nm) was proposed by Toomjeen and co-workers [71]. Negatively charged anti-8-oxo-dG−aptamer interacted with positively charged CA@AuNPs, enabling aggregation-tuned color change from red to blue. The aggregation was disrupted during the recognition of 8-oxo-dG, reversing the color change. An in-depth theoretical insight into the mechanism has been proposed, but it falls short in detailed biomedical applications. Han et al. described using cysteamine-capped NIR-emissive Tin sulfide quantum dots (CA@SnSQDs; size = 16.7 ± 3.2 nm) for HT29 cell imaging studies [72]. Streptavidin (SA)-linked MPA-neutralized SnSQDs were prepared through cysteamine cross-linking to achieve NIR emission at 830 nm. Although this work appears promising, extensive investigation is needed to validate its biomedical applications.

Basiruddin et al. used cysteamine-modified maltose to cap silver clusters (AgCs; size = 1.8 nm) to detect specific glycoprotein concanavalin-A (Con-A, which activates T-cells and plays a key role in lymphocyte studies) [73]. When AgCs interacted with Con-A, the luminescence at 655 nm changed compared to that with BSA. The manifest research lacks detailed information, extended interference studies, and a clear mechanism. Thus, it can be considered an additional report. A human chorionic gonadotropin (HCG, which is a hormone generated during pregnancy) immunoassay was performed through layer-by-layer assembly of AuNPs and cysteamine on a gold electrode [74]. The gold was deposited on a glass electrode (GE) and then assembled with cysteamine via its -SH group, leaving the -NH_2_ free to interact with anti-hCG and BSA. This formed the anti-hCG/hCG complex, which produced a [Fe(CN)_6_]^3−^/^4−^ signal in differential pulse voltammetry (DPV) analysis. As shown in Figure 3, the DPV current exhibited two linear responses for HCG between 1 pg/mL–0.2 ng/mL and 0.2–60.7 ng/mL, with a LOD of 0.3 pg/mL at pH 7.5. The proposed approach showed over 97% recovery in real-sample analysis, with minimal interference. Therefore, it is recognized as a unique contribution. Nevertheless, extensive research is necessary to advance biomedical development.

Ramos et al. advocated the electrochemical synthesis of CA@CDTeQDs (size = 3.0 ± 1.0 nm) for luminescence detection of Resveratrol (a polyphenol present in skin and fruits) [75]. The PL emission of QDs at 575 nm was quenched by Resveratrol, with a linear regression of 3.25–75 µg/mL, and a LOD of 0.97 µg/mL. Notably, the recovery of Resveratrol in wine samples was >97% with <5% RSD values. This research falls short of biomedical analysis. So, it is considered an additional report. Modification of GCEs with CA@AuNPs and poly(amidoamine) dendrimers (PAMAM; generation 4.5) has been reported for uric acid (UA) detection without AA interference [76]. At pH 2, GCE/CA@AuNPs/PAMAM to UA showed a linear response from 0.03 to 0.25 mg/dL, with a LOD and LOQ of 1.7 × 10^−4^ mg/dL and 5.8 × 10^−4^ mg/dL, respectively. The cysteamine conjugate of the electrode stabilized the structure. Human serum samples validate the proposed electrochemical approach. Still, much support is required for biomedical approval.

Hernandez et al. fabricated a screen-printed carbon electrode (SPCE), which has been functionalized with AuNPs-cysteamine SAMs for Urate oxidase (Uox)-derived enzymatic UA detection [77]. Over a buffer-treated SPCE, the AuNPs were electrodeposited, followed by the formation of SAMs using cysteamine via its free -SH and Uox immobilization, affording the final biosensor structure of PTSPCE/AuNPs/CA-SAM/Uox. For UA, the biosensor exemplifies a linear amperometric response from 100 to 1000 µM with a LOD and sensitivity of 4.59 µM and 6.622 nA/μM, as shown in Figure 4. The >100% recovery and <6% RSD values in saliva and urine analysis justify its real-world application. Cysteamine cross-links the AuNPs and Uox, which is crucial in this biosensor fabrication. Yet, further investigation is required for biomedical applications.

Guo et al. utilized the cysteamine-capped CdTeQDs-Ce^4+^ complex for AA’s fluorescence “turn-on” detection [78]. The photoluminescence (PL) intensity of positively charged CA@CdTeQDs (3–5 nm, ζ = +28.16 mV, and Φ_F_ = 10.8%) at 570 nm was quenched by Ce^4+^ via PET between them. During the interaction of AA, the Ce^4+^ was reduced to Ce^3+^, and electrostatic repulsion occured to afford the “turn-on” response. A linear range of CA@CdTeQDs-Ce^4+^ to AA was observed from 33 nM to 33 µM, with an appraised LOD of 11 nM. Time-resolved photoluminescence (TRPL) studies have suggested static quenching between CA@CdTeQDs and Ce^4+^. Human plasma sample analysis displayed >99% recovery and ≤3% RSD values. This is good research, but it falls short in terms of biomedical applications, such as in vitro and in vivo imaging studies. A chiral enzymatic sensor for glucose was developed by mixing CA@CdSQDs with D-pencillamine (DPA) [79]. The chirality of DPA affords a circular dichroism (CD) response during glucose detection. The reaction of glucose oxidase (GOx) and glucose generates H_2_O_2_, which destroys the chirality of DPA/CA@CdSQDs (size = 3.8 ± 0.5 nm), resulting in a CD spectral response. A linear CD response for glucose was perceived from 50 to 250 μM, with an estimated LOD of 31 μM. Spiked coconut water analysis showed >100% recovery and <5% RSD values. However, extensive research is required for biomedical applications.

Li et al. established the resonance light scattering (RLS)-based detection of fructose bisphosphates (FBPs; essential for glycolysis) using uranyl–salophen-modified CA@AuNPs (U-Sal-CA@AuNPs) [80]. For FBPs, the RLS intensity of U-Sal-CA@AuNPs was enhanced linearly from 2.5 to 75 nM, with a LOD of 0.91 nM. Spiked urine analysis showed a 96.5–103.5% recovery. Further research is needed for biomedical approval. Selvi et al. performed the electrochemical detection of doxorubicin (DOX, an anti-cancer drug) using glutathione (GSH) and cystamine-functionalized core–shell molybdenum sulfide nanoparticles (L-GSH–CA@MoS_2_NPs; size = 20–150 nm) [81]. Linear ranges of DOX were found between 0.1–78.3 µM and 98.3–1218 μM, with a LOD of 31 nM. Blood serum analysis justifies this application. Here, cystamine is in a dimeric form of cysteamine; hence, it is accounted as an additional report. CA@CdSQDs detect D-and L-penicillamine (DPA and LPA) via CD spectral studies [82]. Among all amino acid interferences, the CD responses of DPA and LPA were high in the presence of CA@CdSQDs. DPA and LPA have a linear range of 1 to 35 µM with LODs of 0.49 µM and 0.74 µM, respectively. The urine analysis defines its real-time applications. As detailed earlier, the author has extended this research to identify glucose [79]. However, neither report has approved biomedical devices. Therefore, additional research is required for commercialization.

A biosensor was developed with a combination of a cysteamine-modified gold (CA@Au) electrode, 3-mercaptopropionic acid-capped palladium telluride quantum dots (3-MPA@PdTeQDs; size = 4.7 nm), and an enzyme known as heme-thiolate cytochrome P450-3A4 (CYP3A4) for the detection of Indinavir (an anti-viral drug) [83]. The current measured from cyclic voltammogram (CV) studies showed a linear response of 0.3 to 7 ng/mL and a 2.3 pg/mL LOD. This work looks ordinary without any information on the role of cysteamine and in-depth biomedical applications. Gukowsky et al. proposed the colorimetric sensing of Gentamicin (an antibiotic drug) using CA@AuNPs [84]. In the presence of Gentamicin, the SPR peak at 520 nm was diminished with the consecutive appearance of a new peak at 680 nm. The probe has high selectivity among other interferences, with a linear regression of 0–200 nM and a LOD of 50 nM. This work was demonstrated using milk samples; however, extensive research may justify its biomedical applications. 

A direct competitive chemiluminescence immunoassay (CLIA) of chloramphenicol (CAP) was performed by cysteamine and 11-mercaptosuccinic acid-modified gold-coated magnetic particles (Au-MNPs) [85]. The Au-MNPs were conjugated with CAP succinate and CAP base using NHS and NDC. The NSP-DMAE-NHS-labeled antibody was used as a signaling tracer for CAP. While the NSP-DMAE-NHS-labeled antibody interacted with CAP, the CAP base and CAP succinate methods showed linear CLIA responses with LODs of 1 pg/mL and 6 pg/mL, respectively. This method has high sensitivity to CAP but possesses complications in fabrication. There are no valid biomedical applications; hence, it is noted as regular research. Inhibition of FRET between DPA-capped AgInS_2_QDs (size = 4 nm) and CA@AuNPs (size = 107.3 ± 2.2 nm) was pronounced to detect atenolol (a hypertensive medicine) [86]. The PL intensity at 562 nm was quenched due to FRET and restored during the addition of atenolol. Multiple hydrogen bonding between CA@AuNPs and atenolol caused aggregation, which restored the emission of AgInS_2_ QDs. A linear relationship was observed from 1.02 to 11.22 mg/L, with a LOD of 1.05 mg/L. Recovery lies between 97.4% and 104.3% in pharmaceutical formulations with <3% RSD values. The results are impressive; hence, this tactic can be extended for biomedical interrogations.

Kang et al. detailed the colorimetric recognition of clenbuterol (CLB, a drug used in bronchial asthma treatment and an illicit drug used in the feeds of pigs and cattle) using CA@AuNPs (size = 10 nm and ζ = −42.9 mV) [87]. The SPR peak at 525 nm decreased as a new peak appeared at 675 nm, corresponding to the color change from red to blue. The linear range of CLB exists from 50 nM to 1 µM, with a LOD of 50 nM. Blood sample analysis validated this method. Still, there is a long way to go for biomedical device fabrication. Zhang and co-workers revealed the colorimetric and smartphone app-based dual-mode recognition of bongkrekic acid (BA; a respiratory toxin) using CA@AuNPs (size = 38 nm and ζ = +36.8 mV) [88]. For BA, the aggregation-induced colorimetric (red to violet (A_650_/A_524_)) and app-based color responses had linear ranges of 0.18 µM–1.64 μM and 0.10 μM–1.44 μM, with LODs of 3.43 nM and 12.33 nM, respectively. Urine, serum, and tremella analysis showed >89% recovery and <9% RSD values. Based on the smartphone app application tactic, this work could be applied for extensive biomedical monitoring of BA.

Rastogi et al. suggested a colorimetric strategy to detect oxytocin (a hormone and neuropeptide) using CA@AuNPs (size = 8.50 ± 1.41 nm) [89,90]. The RGB study illustrates a linear G/R response for oxytocin from 0.8 to 4.8 IU (IU = International Units), with a corresponding LOD and LOQ of 0.74 IU and 2.26 IU, as shown in Figure 5 [90]. Vegetable-based real analysis demonstrated the validity of the proposed tactic. Yet, additional research may boost its application in food and biomedical research.

Screening of aflatoxin (a carcinogenic substance) was defined by using CA@ZnSQDs and CA@CdSQDs (size = 1–10 nm) via fluorescence response [91,92]. Both works detected the aflatoxin at a LOD of 0.05 ppb (ppb = parts per billion). However, neither of them provided details on real-time applications. Therefore, they are counted as additional reports. Colorimetric identification of fumonisin B1 (mycotoxin derived from Fusarium fungi) using CA@AuNPs (size = 13.6 ± 0.1 nm) via hydrolysis-facilitated aggregation has been reported by Chotchuang and co-workers [93]. The color changed from red to violet with a linear absorbance at 645/520 nm from 2 to 8 μg/kg (kg = kilograms; 1 × 10^3^ g) and a LOD of 0.90 μg/kg. Spiked recovery in corn samples showed 93 to 99% recovery. This is a nice innovation, but extensive research is required for validation.

### Criticism of Biomedical Monitoring Utilities

Use of cysteamine-functionalized nanomaterials has detailed criticisms, which require a careful lookout for future innovations: (1) The majority of CA-functionalized nanomaterials for electrochemical monitoring of biomedical analytes require complicated fabrication steps. Thus, using such tactics may be costly and requires careful optimization. How can their cost become reliable for developing countries? (2) The reliability of many methods is validated mainly by in vitro or solution-based analysis. Therefore, it is not possible to believe the dependability of those reports in diverse environments. (3) The colorimetric stability of proposed strategies for biomedical/biomolecule analytes in diverse environments is not possible. Then, how are they reliable and transportable under diverse environmental conditions? (4) Many cysteamine-functionalized nanomaterials fall short in the details of the role of cysteamine, PLQY, dependable mechanisms, etc. Thus, without such details, the results remain untrustworthy. 

## 3. Environmental Monitoring Applications

Similarly to biomedical screening, the CA-functionalized nanomaterials were pronounced in the detection of environmental contaminants such as metal ions, anions, pesticides, herbicides, nitroaromatics, dyes, and other toxic species, as detailed in this section.

### 3.1. Metal Ions Detection

The detection of toxic Hg^2+^ ions was described using ssDNA oligonucleotides and CA@AuNPs via ECL response [94]. Due to the T-Hg^2+−^T (T = Thymine) complex, the linear ECL response lies from 0.05 to 100 nM, with an estimated LOD of 0.05 nM. This detailed research is an initiative report. Use of CA@AuNPs (size = 13.0 ± 2 nm) for colorimetric sensing (red to blue) of Hg^2+^ and melamine (a milk contaminant) through NH_2_-Hg^2+^-NH_2_ and NH_2–_melamine–NH_2_ stimulated aggregation/assembly has been explored in detail [95]. Linear ranges for Hg^2+^ and melamine were observed between 0.05–3 µM (at A_650_/A_520_) and 0.08–1.6 µM (at A_650_/A_520_), with respective LODs of 30 nM and 80 nM. The Hg^2+^ and melamine in spiked real and milk powders recovered 90 to 96.7% and 98 to 102%, respectively. This research innovated the use of CA@AuNPs for environmental screening. Naked-eye and colorimetric sensing of Hg^2+^ ions using CA@AgNPs (size = 14−26 nm) via Ag-Hg nanoalloy formation was discovered by Bhattacharjee and co-workers [96]. From 0 to 13.6 nM, the UV–visible peak at 402 nm was linearly diminished, with an appraised LOD of 275 pM. Conversely, the naked-eye response (yellow to colorless) had a LOD of 2.73 nM. The optimal pH values for the sensor were detailed as 3 to 8.2. Real water analysis yielded 96−102% recovery. This is state-of-the-art research, but extensive interrogations are required for valid device-based applications.

AuNPs’ capability to recognize the aptamers’ conformational changes under high ionic strength was applied to design CA@CdTe QDs/AuNPs/NaCl/aptamer system for luminescent detection of Hg^2+^ ions [97]. Thymine of aptamers attracts the Hg^2+^ to form the T-Hg^2+^-T complex, resulting in NaCl-induced AuNP aggregation and restoration of CDTeQDs’ fluorescent enhancement at 522 nm. The linear range for Hg^2+^ lies between 50 pM and 1 nM, with a calculated LOD of 2.5 pM. Tab and Lake water samples display >95% recovery and <5% RSD values. This work is a nice one. However, complications in sensor fabrication raise questions about its reliability in diverse environments. Advancing silver ion (Ag^+^)-modulated CA@CdSQDs (size = 3.3 ± 0.8 nm) for fluorescent sensing of Hg^2+^ ions has been carried out by Ngeontae’s research unit [98]. While Ag^+^ interacts with CA@CdSQDs, the PL at 545 nm was enhanced due to a decrease in nanocrystal defects. During the recognition of Hg^2+^, the PL emission was quenched with a red shift from 545 nm to 550 nm. Metallophilic attraction between Hg^2+^ and Ag^+^ ions may result in fluorescence enhancement. The linear range of Hg^2+^ exists between 0.1 µM and 3.5 μM with an appraised LOD of 90 nM. Recovery analysis in drinking water validated this tactic. However, additional support is still needed.

Chen et al. advanced a portable RGB device using CA@CdTeQDs (size = 2–4 nm) to detect Hg^2+^ ions [99]. The CA@CdTeQDs exhibit a linear fluorescence quenching at 564 nm from 5 to 300 nM, with a LOD of 1 nM, as shown in Figure 6. In addition, the RGB device demonstrates a linear response from 5 to 200 nM. Standard addition of Hg^2+^ in tap water yields over 97% recovery with less than 14% RSD values. At pH 7.54, the -NH_2_ of cysteamine binds with Hg^2+^ to afford a PL quenching response. This research lacks the detail of static or dynamic quenching phenomena.

Conjugation of positively charged core–shell CA@CdTe/CdSQDs (size = 7 nm) with the negatively charged T-rich aptamers for PL-based detection of Hg^2+^ has been demonstrated by Rezaei’s research group [100]. Interaction of T-rich aptamers quenches the PL emission at 525 nm and regenerates the emission during Hg^2+^ addition. This might be due to the T-Hg^2+^-T formation, resulting in the original emission of CA@CdTe/CdSQDs. Linear emission enhancement by Hg^2+^ addition was found from 0.5 nM to 1 µM, with a LOD of 80 pM. Though real water analysis has validated this tactic, additional data are required to proceed further. Incubation of a CA@CdS QDs-T/GCE in luminol@Au-Cys-T (Cys = Cysteine and T = thymine) nanoprobes with Hg^2+^ resulted in a linear ECL response from 5 pM to 2.5 nM, with a LOD of 2 pM [101]. Due to the T-Hg^2+^-T interaction, ECL intensity was enhanced for Hg^2+^ ions. Spiked tap, river, and lake water investigations revealed >90% recovery and <7% RSD values. Extensive research is mandatory for the fabrication of portable devices.

Shellaiah et al. synthesized cysteamine-functionalized nanodiamonds (CA@NDs; size = 20–250 nm, ζ = +10.24 mV, and Φ_F_ = 13%) for PL-based detection of Hg^2+^ ions [102]. The CA@NDs have free -SH to bind with Hg^2+^ and aggregate to exhibit an intense PL peak shift from 440 to 463 nm, as shown in Figure 7. A 0–100 µM linear regression was appraised for Hg^2+^, with a LOD of 153 nM. Agglomeration-induced partial graphitization and nanowire-like assembly were proposed as a mechanism. This work was demonstrated by HeLa cellular imaging. However, due to the formation of nanowires at diverse pH values [103], the detailed sensor cannot be used in a wide range of pH conditions. Olenin et al. described the spectrophotometric detection of Hg^2+^ using CA@AgNPs by Ag-Hg nanoalloy formation [104]. The marked research holds similarities to the earlier report [96]; hence, it cannot be explored further in this review.

Ngeontae’s research group utilized CA@CdSQDs (size = 3.6 ± 1.2 nm) for “turn-on” recognition of Ag^+^ ions by modulating radiative and non-radiative recombination to reduce the nanocrystal defects [105]. Adding Ag^+^ between 0.1 µM and 1.5 µM, the PL emission of CA@CdSQDs at 525 nm was enhanced, with LOD and LOQ values of 68 nM and 0.2 µM, respectively. The author applied this method to quantify free Ag^+^ in AgNPs solution. It could be rated as an initiative work, and the author wisely extended it to PL-quenching-based identification of Hg^2+^ ions, as delineated earlier [98]. Real-time applications are required to validate this tactic. Ensafi et al. advanced an aptasensor using cysteamine-stabilized core–shell CdTe/ZnSQDs (size = 3–4 nm) to detect As^3+^ ions [106]. In fact, the aptamer induced the aggregation of QDs, resulting in PL quenching at 530 nm. During the sensor studies, the aptamer bound to As^3+^ and underwent de-aggregation of QDs, resulting in PL enhancement at 530 nm. The dynamic linear range for As^3+^ was found between 10 pM and 1 µM, with a LOD of 1.3 pM. This study falls short of extensive real-time investigation. Thus, additional research is in demand for endorsement.

Cystamine (dimeric form of cysteamine)- and glycidyl methacrylate (GMA)-modified magnetic nanoparticles, “Fe_3_O_4_@SiO_2_@GMA-CANPs”, were produced via click reaction and applied for magnetic solid-phase extraction (MSPE) of As^3+^ ions [107]. The proposed MSPE-ICP-OES technique extracts the As^3+^ linearly between 2.7 nM and 1.3 µM and possesses a LOD of 667 pM. Contaminated water analysis validated this tactic. Based on the results, this research can be rated exceptional in environmental monitoring and remediation. Yadav et al. cross-linked the DL-glutaraldehyde with CA@AuNPs to afford DL-G-CA@AuNPs for colorimetric detection of Cd^2+^ ions via aggregation [108]. The SPR peak shifted from 520 nm to 736 nm, relevant to the color change from red to blue. Linear regression of Cd^2+^ lies from 0.05 to 500 μM, with an estimated LOD of 21 nM. The real-water analysis by this method revealed ≥98% recovery with <3% RSD values. This work followed the standard AuNPs-based colorimetric strategy. Therefore, it can be considered an additional report.

Boonmee et al. developed cysteamine-capped copper nanoclusters (CA@AuNCs; 2.3 ± 0.5 nm) for “turn-on” detection of Al^3+^ ions [109]. During Al^3+^ detection, the emission of CA@AuNPs at 380 nm and 430 nm was enhanced linearly from 1 to 7 µM with an appraised LOD of 26.7 nM. It has been stated that aggregation of NCs may be responsible for the Al^3+^ sensor. The method showed 91–101% recovery in drinking water, with <4% RSD values. The PL “turn-on” results are not unique, and this research falls short of exceptional PLQY changes. Therefore, it cannot be recommended for commercial purposes. Shervedani et al. functionalized the cysteamine SAMs over gold with ethylenediaminetetraacetic acid (EDTA) to attain an EDTA SAM electrode for electrochemical sensing of Cu^2+^ and Pb^2+^ ions [110]. Both Cu^2+^ and Pb^2+^ ions showed a linear SWV response from 10 nM to 1 µM, with LODs of 72 pM. This work was published 18 years ago; hence, it can be noted as preliminary research.

Wang et al. constructed a FRET system consisting of negatively charged 11-mercaptoundecanoic acid-capped AuNPs (MUA@AuNPs; size = 14 nm and ζ = −44.57 mV) and positively charged CA@CdTeQDs (size = 3–4 nm and ζ = + 21.96 mV) for the detection of Pb^2+^ ions [111]. The emission of CA@CdTeQDs at 560 nm was quenched during the conjugation of MUA@AuNPs due to FRET between them. In the presence of Pb^2+^, the FRET was disturbed via the coordination between Pb^2+^ and MUA@AuNPs, resulting in aggregation and color change from red to blue. The SPR peak of AuNPs was shifted from 523 nm to > 600 nm. Similarly, the RLS and PL intensities at 400 nm and 560 nm showed a “turn-on” response. Linear regression ranged from 1.06 µM to 21.8 µM, with a LOD of 145 nM. Additional research on real-time applications is essential for authentication. Madrakian and co-workers employed CA@CdS hollow nanospheres (CA@CdSHNSs; diameter = 110 ± 15 nm) to quantify Cd^2+^ and Pb^2+^ by the RLS technique [112]. While pH was maintained at > 4.8, the -NH_3_^+^ of the cysteamine layer over CA@CdSHNSs deprotonated, which attracted Cd^2+^ and Pb^2+^ to afford the RLS response at 445 nm, as shown in Figure 8. The RLS-based linear response of CA@CdSHNSs to Cd^2+^ and Pb^2+^ was established as 445 nM–44.5 µM and 48 nM–19.3 µM, with LODs of 89 nM and 43 nM, respectively. This method showed >95% recovery in tap and river water samples, which is impressive. But a detailed investigation with interference and environmental samples may validate its reliability.

By integrating carboxyl-modified upconversion nanoparticles (UCNPs), gold nanorods (AuNRs), and cysteamine, researchers have developed a highly effective FRET-based weak fluorescent sensor probe, called UCNPs-Cys-GNRs, to detect Pb^2+^ in food samples [113]. CA@AuNRs were assembled over UCNPs, and FRET occurred between AuNRs and UCNPs, resulting in weak emission. During Pb^2+^ sensing, the FRET was disrupted by coordination with CA. As a result, a linear “turn-on” emission at 547 nm was observed for Pb^2+^ concentrations from 1 µM to 100 µM, with a LOD of 0.5 µM. This approach showed >95% recoveries and <8% RSD values in matcha samples. Further research is needed to determine the stability of the FRET system under various environmental conditions. Hydrothermally synthesized carbon dots (CDs; size = 3–9 nm) from jeera (Cumin seeds: Cuminum cyminum) have been capped with cysteamine using EDC and NHS and applied to detect Cr^6+^ ions [114]. For Cr^6+^ ions, CA@CDs perceive a linear PL quenching response from 1 µM to 10 µM, with a LOD of 1.57 µM. The author defined the applications in MCF-7 cancer cell lines. Nevertheless, this work lacks information on PLQY and the static/dynamic quenching mechanism. Therefore, it can be noted as an additional report in environmental screening.

Shellaiah and co-workers synthesized cysteamine-functionalized gold–copper nanoclusters (CA@Au-Cu NCs; size = 4.6 ± 3.3 nm, Φ_F_ = 18%, and ζ = −12.76 mV) for selective detection of Cr^6+^ and dopamine (DA) [115]. For Cr^6+^ and DA, the linear measurement ranges were identified as 0.2 to 100 μM and 0.4 to 250 μM, with LODs of 80 nM and 135 nM, respectively. This research has been validated by spiked water analysis (>98% recovery and <4% RSD) and cellular imaging in Raw 264.7 cell lines. But additional investigations are needed to clarify the proposed static quenching mechanism. Hydrothermally synthesized CDs from Ruellia simplex flowers were engaged in detecting Cr^6+^ ions [116]. Adding Cr^6+^ to the unmodified CDs (RS-CDs; size = 3.17 ± 0.6 nm) and cysteamine-modified CDs (RS-CA@CDs; size = 3.09 ± 0.6 nm) quenched the PL emission at 420 nm and 422 nm, respectively. The linear response of RS-CDs and RS-CA@CDs with Cr^6+^ was established as 3–100 µM, with LODs of 1.28 µM and 1.58 µM, respectively. An inner-filter effect (IFE) was proposed as a mechanism. However, this supplementary application lacks validation. Therefore, it cannot be recommended as an effective tactic for Cr^6+^ quantification.

Li et al. consumed CA@AuNPs for colorimetric recognition of Cu^2+^ ions [117]. The color was changed from red to blue, with SPR linear regression of 0–10 µM and a LOD of 0.4 µM. Applying this tactic in tap, river, and lake water analysis showed recovery between 97.5% and 101.5%. This initiative project needs additional research to lower the LOD and improve authentication. Ngamdee et al. described circular dichroism (CD)-based detection of Cu^2+^ by using the mixture of achiral CA@CdSQDs (size = 1.8 ± 0.3 nm) with D-penicillamine (DPA; 100 μM) [118]. DPA-CA@CdSQDs showed a linear CD response from 0.5 to 2.25 µM, with a LOD of 0.34 µM. The proposed method displayed 102–114% recoveries, with <6% RSD values in drinking water. Therefore, it can be noted as an exceptional innovation, and further research may lead to commercialization. Pal et al. defined the PL quenching of CA@CdTeQDs at 576 nm by Co(III) complex through electrostatic assembly [119]. Static quenching was established in this research based on multiple time-resolved investigations. This work can be extended to Co(III) monitoring in complexes. Tai et al. used CA@CuNCs (size = 1–5 nm) to detect Fe^3+^ and I^−^ ions [120]. As shown in Figure 9, CA@CuNCs aggregated with Fe^3+^ and I^−^ ions to afford the PL quenching response. The linear range for Fe^3+^ and I^−^ was identified as 0–1000 µM and 0–10 mM, with assessed LODs of 423 nM and 2.02 µM, respectively. Bioimaging, the strip method, and human urine analysis validated this method. Nevertheless, information like static/dynamic quenching and electron transport between NCs and analytes has not been updated. Thus, it is considered an additional report. Conjugation of cysteamine-functionalized nanodiamonds (NDC) with AuNPs affords NDC@AuNPs, which shows a reversible colorimetric selectivity to Cr^3+^ ions at pH 6 [121]. For Cr^3+^, NDC@AuNPs (size = 18.3 ± 4.8 nm and ζ = −29.80 mV) possess a linear response from 10 to 400 nM with a color change from red to purple. SPR analysis estimated a LOD of 0.236 ± 0.005 nM. The colorimetric response was assigned to particle aggregation, and it became reversible with EDTA up to four cycles. Relevant to inductively coupled plasma–mass spectrometry (ICP-MS), this tactic has been validated by spiked tap and lake water studies, which showed 96–110% recovery with <3% RSD. Thus, it can be noted as a nice innovation in nanosensors. 

### 3.2. Anions Quantification

Zhang et al. proposed an anti-aggregation of CA@AuNPs to detect thiocyanate (SCN^−^) in the presence of N, N-dimethyl-1-naphthylamine (represented “2N”) [122]. The 2N interacts with the protonated free amine (-NH_3_^+^) of CA@AuNPs, leading to aggregation-induced color change from red to purple. While treating the above system with anions, SCN^−^ induces anti-aggregation, which restores the red color. Due to the strong coordinating ability of SCN^−^ to -NH_2_ of cysteamine, the 2N was replaced to deliver the SPR (A_680 nm_/A_520 nm_) and colorimetric responses. The proposed method had a LOD of 0.2 µM for SCN^−^ ions. Spiked urine sample analysis showed >94% recoveries and <7% RSD values. This work lacks transmission electron microscope (TEM) and zeta size data; thus, it looks ordinary. Noipa and co-workers pronounced Cu^2+^ modulation to quench the emission of CA@CdSQDs (size = 4.16 ± 0.86 nm), followed by the addition of CN^−^ ions to restore PL intensity [123]. At 557 nm, the emission was linearly recovered from 2.5 to 20 µM, with a LOD and LOQ of 1.13 µM and 3.23 µM, respectively. During the addition of CN^−^, Cu^2+^ tends to form Cu(CN)_2_, resulting in PL recovery. Drinking water studies have shown 94–110% recovery and ≤6% RSD values. This research is an initiative and can be extended for commercial purposes.

Elsner reaction-facilitated colorimetric detection of CN^−^ ions, using CA@AuNPs (size = 23 ± 2.5 nm; ζ = +45.41 mV), was discussed [124]. The CN^−^ ions replace the CA on the AuNPs surface to form 4[Au(CN)_2_]^−^, resulting in aggregation of NPs with a red-to-weak blue color change. Linear regression of CN^−^ at 525 nm was observed between 1.5 µM and 30 µM with a LOD of 128 nM, as shown in Figure 10. A smartphone-based RGB sensing platform displayed 91.9–96.7% recovery, with <4% RSD values. Use of CA@AuNPs as a photocatalyst for the reduction of 4-Nitrophenol to 4-Aminophenol was also demonstrated in this report. This is an interesting work to be extended for commercial sensing device fabrication.

Zhao et al. inhibited the peroxidase-like activity of CA@AuNPs (Size = 25 nm) to deliver the colorimetric sensing of sulfate (SO_4_^2−^) ions [125]. In the presence of SO_4_^2−^ ions, the CA@AuNPs were coupled to a 3,3′,5,5′-tetramethylbenzidine (TMB)–H_2_O_2_ peroxidase reaction to afford a color change from strong blue to faded blue. The relative absorbance peak at 652 nm showed a linear response from 0.2 to 4 µM and an evaluated LOD of 0.16 µM. Spiked real water sample analysis displayed 92–104% recovery, with < 5% RSD values. This research is impressive, but it lacks a commercial scope. By using CA@AuNPs, the surface-enhanced Raman scattering (SERS)-based detection of perchlorate (ClO_4_^−^) has been defined with a LOD of 5 µM [126]. Only a few interferences, such as NO_3_^−^, Cl^−^, SO_4_^2−^, and PO_4_^3−^, were used in this study. Later, it was acknowledged that CA@AuNPs (size = 10–30 nm) have high selectivity to NO_3_^−^ [127]. Therefore, this research can be noted as an additional report due to the lack of clarification. The boiling of CA@AuNPs in the presence/absence of sodium chloride reveals a color change from red to purple [127]. This tactic can detect NO_3_^−^ below 0.6 µM. However, it lacks details on interferences, linear range, and real applications. Thus, it can be counted as an additional report. Sim et al. detailed the Mn-doped cysteamine-capped ZnS nanocrystals (Mn-CA@ZnSNCs; size = 278 nm in water; ζ = +10.89 mV) for the detection of nitrite (NO_2_^−^) ions [128]. During the recognition of NO_2_^−^, the PL emission at 550 nm (Φ_F_ = 0.546) was quenched linearly from 0 to 100 µM at a rate constant of k = 2.60 ×10^5^ M^−1^. This work lacks detailed information on the proposed IFE mechanism and real applications.

### 3.3. Nitroaromatics Sensing

A hybrid CdTeQD (size = 3.7 nm) has been used to detect trinitrotoluene (TNT) and picric acid (PA) through a PL quenching response [129]. For PA and TNT, the PL emission around 610 nm was linearly quenched from 2 to 15.3 µM and from 0.4 to 15.3 µM, with LODs of 50 nM and 60 nM, respectively. This preliminary work lacks detailed mechanistic studies and real-time applications. Gao et al. constructed a FRET system by compositing CA@AuNPs and mercaptopropionic acid-capped CdTe QDs for colorimetric and fluorometric monitoring of TNT [130]. The GNPs@QDs have a self-assembled structure due to coulomb attraction, leading to SPR and quenched PL peaks located at 538 nm and 540 nm, respectively. During TNT recognition, CA@AuNPs undergo aggregation with a color change from purple–pink to yellow, and the yellow emission of CdTeQDs emerges, as shown in Figure 11. Related to the above observation, the UV–visible peaks at 543 nm and 795 nm show a linear response from 50 nM to 200 µM and a LOD of 16 nM. Likewise, the PL titration displays a linear enhancement at 540 from 1 nM to 5 µM with a LOD of 0.24 nM. Using SERS-based detection, this material shows a LOD of 3.2 fM (fM = femtomole; 10–15 M). Concerning the lower LODs across multiple techniques, negligible interferences, and applications in environmental and biological specimens (holds > 83% recovery), this research can be noted as exceptional for detecting TNT in commercial devices.

Kumar et al. fabricated a nanosensor by compositing cysteamine-capped cadmium selenide quantum dots (CA@CdSeQDs; size = 4.3 nm) anchored over graphene xerogel (surface area = 238.3 g/m^2^) to detect TNT via static PL quenching at 499 nm [131]. The author found a linear range between 0 µM and 311.4 µM, with a LOD of 9.7 µM. This research lacks details regarding interference, mechanism, and real-time applications; hence, it is noted as an additional report. Cost-effective paper-based voltammetry detection of TNT has been performed using cysteamine-linked Fe_3_O_4_@Au NPs (size is ≥ 100 nm) [132]. The core–shell NPs showed high selectivity to TNT among other interferences, within a linear range of 2 nM–10 µM and a LOD of 0.5 nM. Real sample analysis demonstrated 98–103% recovery with high reproducibility. Reduction of the nitro (NO_2_) group into an amine was proposed as a mechanism. Concerning its cost-effectiveness, portability, easy fabrication, reproducibility, and applicability, it can be forwarded for commercialization. Nevertheless, an electrochemical station is required in this tactic.

### 3.4. Screening of Pesticides and Herbicides

Hydroquinone and catechol (HQ and CT, present in pesticides and pharmaceutical waste) detection was illustrated using CA@AgNPs (size = 4 nm) and a single-walled carbon nanotube-modified GCE (CA@AgNPs-CNTs/GCE) [133]. Linear DPV responses for HQ and CT were established as 80 nM–200 µM and 0.2–280 µM, with estimated LODs of 10 nM and 40 nM, respectively. River and cream investigations demonstrated real-time recovery (>74%). Still, additional research is required to enhance the recovery rate. Hussain et al. described SERS-based sensing of oxamyl and thiacloprid pesticides in milk using cysteamine-capped silver-coated gold nanoparticles (Au-CA@AgNPs; size = 28 nm) [134]. SERS signals at 679 cm^−1^ and 1095 cm^−1^ were assigned to oxamyl and thiacloprid, respectively. In milk, linear regression was found at 2.3–46 µM and 2–40 µM, with LODs of 141 nM and 91 nM for oxamyl and thiacloprid, respectively. The recovery of oxamyl and thiacloprid in milk was found to be >98%, which is impressive. Fabrication of a portable device using this technique may boost its commercial value.

Liu et al. used CA@AuNPs (size = 13 nm) for colorimetric sensing of atrazine in detail [135]. The SPR peak at 523 nm was shifted to 640 nm, with a color change from wine red to blue. Linear regression of atrazine ranged between 0.033 µg/g and 6.67 µg/g with a LOD of 0.0165 µg/g. Rice sample analysis showed 83–92% recovery, with <4% RSD values. Atrazine coordination with -NH_3_^+^ of cysteamine induced aggregation to envisage a colorimetric response. This is innovative research, but further investigations are mandatory for endorsement. Professor Sun’s research group constructed a FRET system using positively charged CA@AuNPs (size = 13 nm and ζ = +15.6 mV) and negatively charged thioglycolic acid-capped CdTe quantum dots (TGA-CdTe QDs; size = 1.9 nm and ζ = –35.9 mV) via electrostatic interactions, which were applied for glyphosate detection [136]. While detecting glyphosate (possessing the dual negative charge), it binds to CA@AuNPs to release QD dispersion and emission at 532 nm, as schematically illustrated in Figure 12. Fluorescent recovery at 532 nm showed a linear response from 0.02 to 2.0 μg/kg, with a LOD of 9.8 ng/kg. Due to the glyphosate-stimulated aggregation of CA@AuNPs, the color varied from red to blue, relevant to the SPR response at 528 nm. Recoveries in apples were estimated between 88.5% and 102.6%. However, extensive investigations are required for validation.

Long-period grating-coated CA@AuNPs were pronounced for the detection of glyphosate [137]. Nevertheless, this work lacks details on the mechanism, interferences, and applications. Therefore, it is counted as an extra report. Zheng et al. described the colorimetric sensing of glyphosate using CA@AuNPs (size = 30 nm) [138]. An electrostatic binding-stimulated aggregation occurred between glyphosate and -NH_3_^+^ of CA@AuNPs, leading to a red to blue color variation. A UV–visible absorbance ratio at A_650_/A_524_ linearly varied from 0.5 to 7µM, with a LOD of 58.8 nM. The proposed method achieved more than 95% recovery in tap water samples, marking a significant advancement in research. Professor He’s research group demonstrated a SERS-based glyphosate screening in plant tissue using CA@AuNPs (size = 50 nm) [139]. The SERS peaks for glyphosate at 1172 cm^−1^ and 1591 cm^−1^ were linearly enhanced between 1 µg/L and 1 g/L, with a LOD of 26 µg/L. The apple peel, spinach, and corn leaf-based results approve the strategy as a standout one for portable device fabrication. Antibody-functionalized AuNPs and CA@AuNPs were described to detect glyphosate [140]. The CA@AuNPs and anti-glyphosate–AuNPs showed linear ranges of 0.180–3 mg/L and 0.3–20 µg/L, with LODs of 42 µg/L and 0.15 µg/L. This work demonstrates the reliability of anti-glyphosate–AuNPs over CA@AuNPs. Thus, it is not explored further.

An enzymatic colorimetric assay of parathion ethyl (an organophosphate pesticide) has been defined using CA@AuNPs (size = 13.24 nm and ζ = +39.4 ± 2.05 mV) [141]. Step-wise addition of acetylcholinesterase (AChE), parathion ethyl, acetylcholine chloride (ACh), CA@AuNPs, and TMB led to a color change from mild blue to deep blue. In fact, parathion ethyl is a neurotoxic agent that blocks the AChE, which plays a crucial role in delivering sensory signals. The relative absorbance peak at 652 nm has a linearity from 40 to 320 nM and a LOD of 20 nM. This is innovative research, but a time-consuming tactic to achieve a sensor response. Ma et al. demonstrated SERS-based quantification of pentachlorophenol (PCP, which exists in insecticides, fungicides, and environmental contaminants) using CA@AuNPs (size = 17–25 nm) [142]. The positively charged NH_3_^+^ of cysteamine interacts with PCP to afford a SERS response at 649 cm^−1^. Linearity was achieved between 1 nM and 100 µM of PCP, with an estimated LOD of 1 nM. Tap water interrogations showed >95% recovery with a LOD of 1 nM. With respect to stability, selectivity, and reproducibility, this work can be considered exceptional for portable device fabrication. Jiang and Zhan’s research groups detailed the use of CA@AgNPs for SERS-based sensing of PCP [143]. The SERS response at 640 cm^−1^ was linearly enhanced for PCP from 0.5 to 100 μM, with a LOD of 0.2 µM. The NH_3_^+^ of cysteamine electrostatically attracts PCP to induce aggregation and the SERS response. This research lacks real-time applications; thus, additional interrogations are necessary for approval.

### 3.5. Other Environmental Contaminants Detection

Cysteine-functionalized AuNPs detect the marine toxin Saxitoxin (STX, a paralytic shellfish toxin harmful to humans) via SERS response [144], which could be tuned using CA@AuNPs for other environmental contaminants. For instance, the conjugation of graphene QDs (size = 6–11 nm) and CA@AuNPs (size = 23 nm) has been explored for the detection of Erythrosine B (ErB, a food colorant and environmental contaminant) [145]. Here, the -SH and -NH_2_ of cysteamine conjugate to the AuNPs and GQDs, respectively. While adding the ErB to GQDs-CA@AuNPs, the PL intensity around 510 nm was quenched linearly from 1.2 nM to 50 nM, with a LOD of 0.03 nM. Involvement of both static and dynamic quenching, depending on the temperature, has been discovered in detail. This is nice work, but it lacks detailed applications. Since the -OH groups in GQDs play a crucial role in this photocatalytic sensor, it has been noted that both QDs and NPs become essential in fabricating such a sensor. He et al. used the CA@AuNPs as a SERS substrate for acidic pigment detection with a limit of 1 ppm [146]. The NH_3_^+^ of cysteamine electrostatically binds to deliver diverse SERS bands. Detection of dyes like allura red, sunset yellow, lemon yellow, and acid blue has been illustrated by this tactic. However, extensive research is mandatory for specific dye monitoring. Cysteamine-capped gold nano-bipyramids (CA@AuBPs), CA@AuNP-modified attapulgite (Au/ATP NCs), and anthraquinone–cysteamine-functionalized AuNPs over SPEs were demonstrated for weak surficial affinity molecules, H_2_O_2_, and dissolved oxygen (O_2_), respectively [147,148,149]. The CA@AuBPs detects allura red and sunset yellow at a limit of 0.1 ppb and 1 ppb via SERS but lacks specificity [147]. For O_2_, the anthraquinone- and cysteamine-functionalized AuNPs on SPE show a linear response from 0.2 to 6.1 mg/L, with a LOD of 0.131 mg/L [149]. Though the author claims that it is a cost-effective method, complications in fabrication raise questions about its reliability. Additional information on specificity and applicability is required to approve the tactics proposed later [147,148,149].

### 3.6. Critical Views on Environmental Contaminants Detection

As outlined below, several critical issues are associated with using cysteamine-functionalized nanomaterials for ecological monitoring. (1) Using positively charged cysteamine-functionalized nanomaterials for metal ions requires critical evaluation to address the question of ‘How is it possible to coordinate two positively charged species to afford the sensor response?’. (2) The use of CA@AuNPs has been pronounced for the detection of CN^−^ and NO_3_^−^ by adjusting the operable temperature [124,127], thus the reliability of proposed sensors is in question. This requires many critical evaluations. (3) Fabrication of cysteamine-functionalized nanomaterials in electrochemical sensors for metal ions, anions, and other ecological analytes shows complicated steps and requires a portable electrochemical station. How can such research be implemented in developing countries and commercial devices? How could it become cost-effective and operable at places that do not have electricity? These questions must be addressed in detail via critical investigations. (4) While using peroxidase-like photocatalytic colorimetric sensors with cysteamine-capped nanomaterials, the enzyme stability is not trustworthy under harsh environmental conditions, and it has a time-consuming process. How is it helpful to quantify the analytes in diverse environments, and how can we increase their rapid sensing efficacy? (5) The specificity of cysteamine-functionalized nanomaterials via the SERS method for acidic dyes has not been clarified for future researchers. How could it be dependable for environmental monitoring and remediation? 

## 4. Advantages and Limitations

Employing cysteamine-functionalized nanomaterials for biomedical, biomolecular, and ecological analytes has several advantages and a few limitations, as discussed in detail.

### 4.1. Advantages

(1)Due to the presence of free amine (-NH_2_) and thiol (-SH) groups in cysteamine, the functionalization of nanomaterials using any of those groups leaves the other one as a free group to attract specific analytes.(2)By tuning the charge potential in cysteamine-functionalized nanomaterials, both aggregation and anti-aggregation can be achieved during analyte sensing, which is advantageous in delivering distinct responses for analytes.(3)The use of CA@AuNPs for optical reflectometry and colorimetry-based detection of SARS-CoV-2 (COVID-19) [51] has similar advantages to an electrochemical sensor driven by SAM-facilitated immobilization of SARS-CoV-2 virus proteins [150].(4)Cysteamine-functionalized nanomaterials could afford a large surface area with tunable charge potential to attract diversely charged analytes with high selectivity, which is advantageous in electrochemical and SERS-based sensing platforms over other functional nanomaterials and analytical methods [151,152,153].(5)It is highly feasible to conjugate two different nanomaterials to afford composite structures and a FRET system, which is helpful for biomedical and ecological monitoring studies. This is quite similar to the direct conjugation of cysteamine in organic sensory probes [154,155,156].(6)Similarly to organic probes [157,158,159], cysteamine-functionalized nanomaterials can also be employed for multi-hazard detection simultaneously. This is a standout advantage over other techniques.

### 4.2. Limitations

(1)Cysteamine-functionalized nanomaterials’ stability and sensor response at harsh environmental conditions depend on surface potential and optimized results, which could limit the reliability and commercialization of those materials.(2)Many available electrochemical sensors, illustrated by cysteamine-functionalized nanomaterials, require complicated multi-step fabrication procedures, which could restrict their commercialization.(3)SAM-based sensor performance depends on its stability, which may be greater in longer -CH2- chain containing thiols than cysteamine [160]. This could limit the design of such sensors.(4)Enzymatic peroxidase-based sensors using cysteamine-functionalized nanomaterials are found to be a time-consuming process. Similarly, FRET probes require careful optimization of concentration. This could limit their operation in portable sensory devices.(5)Until now, reports using cysteamine-functionalized nanomaterials for biomedical and ecological monitoring wisely avoid the feasible interferences, which are impossible at real-time detection. This could restrict the fabrication of commercial devices without any valid trials.(6)To characterize cysteamine-functionalized nanomaterials at research and commercial levels, costlier instruments, such as atomic force microscopy (AFM), scanning electron microscopy (SEM), transmission electron microscopy (TEM), zeta sizer, X-ray diffraction (XRD), X-ray photoelectron spectroscopy (XPS), etc., are required. This could limit the use of those materials in developing countries.

## 5. Conclusions and Perspectives

This review explored the biomedical, biomolecular, and ecological monitoring utilities of cysteamine-functionalized nanomaterials in research, moving a step closer to commercial purposes. The effective sensing performance of cysteamine-capped nanomaterials, such as nanoparticles, nanoclusters, nanocomposites, and other nanostructures, has been presented with clarification. The involvement of diverse sensory tactics like colorimetry, fluorometry, electrochemical approaches, and SERS has been explained by the researchers. The mechanistic aspects and role of cysteamine in each sensory report have been described for future researchers. However, focusing on the following perspective points could lead to impactful research in this field: (A) The use of cysteamine-functionalized nanomaterials for biomedical monitoring was primarily delivered by electrochemical sensors with complicated multi-step fabrication, which could be simplified to achieve portable state-of-the-art commercial devices. (B) Construction of FRET system using cysteamine-functionalized nanomaterials could be extended with emerging nanomaterials such as MXenes, perovskites, carbon dots, etc. (C) Functionalization of cysteamine mainly involves free thiol (-SH) groups over nanomaterials, which could be overturned to achieve effective detection of Hg^2+^, Pb^2+^, Cd^2+^, and other thiophilic analytes. (D) Insufficient reports on anions and nitroaromatic sensors using cysteamine-functionalized nanomaterials open doors for researchers. By tuning nanostructural features and diverse materials, researchers can address this issue in an impactful way. (E) Cysteamine-functionalized nanomaterial-based peroxidase-mimic sensors appear time-consuming; thus, research focused on attaining a rapid colorimetric response may lead to the generation of intellectual property and portable devices. (F) Contaminant free-injectable and portable microfluidic chips and devices could be fabricated with cysteamine-functionalized nanomaterials to overcome stability/cost challenges [161]. (G) Combinations of Artificial Intelligence (AI) [162], machine learning [163], and density functional theory (DFT) studies may boost the monitoring features of cysteamine-functionalized nanomaterials to deliver state-of-the-art commercial products for day-to-day life.

Apart from those open questions, cysteamine-functionalized nanomaterials hold their merits in biomedical, biomolecular, and ecological monitoring and remediation. Ongoing and upcoming research in this field may undoubtedly address all issues. 

## Figures and Tables

**Figure 1 micromachines-16-01144-f001:**
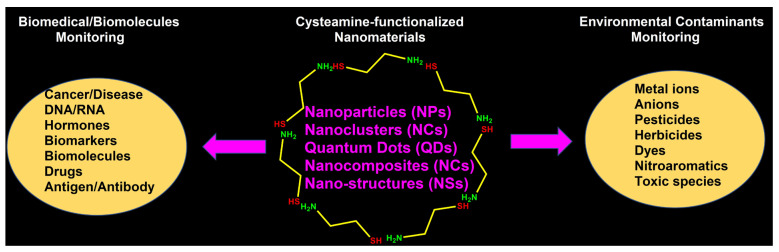
Schematic representation of cysteamine-functionalized nanomaterials for biomedical, biomolecular, and environmental monitoring applications outlined in this review.

**Figure 2 micromachines-16-01144-f002:**
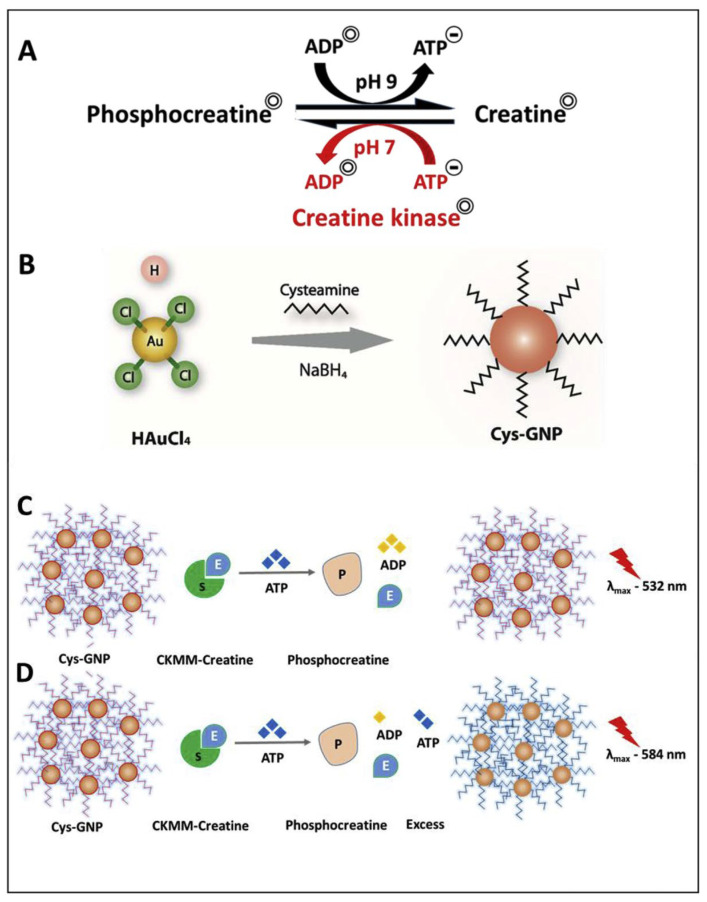
(**A**) shows enzyme activity with creatine as the substrate and ATP as an activator, producing phosphocreatine at physiological pH 7. The reaction can reverse at pH 9. Panel (**B**) illustrates the synthesis of cysteamine-capped gold nanoparticles (Cys-GNP) using sodium borohydride (NaBH_4_) as a reducing agent. Panel (**C**) depicts the initial reaction where an excess of high-concentration CKMM can complete the enzyme-substrate reaction, converting ATP to ADP. Panel (**D**) demonstrates the mechanism of sensing creatine kinase through ATP-induced aggregation of the initial Cys-GNP (red color) into a blue solution with an optical spectral change (red shift) (reproduced with permission from ref. [60]).

**Figure 3 micromachines-16-01144-f003:**
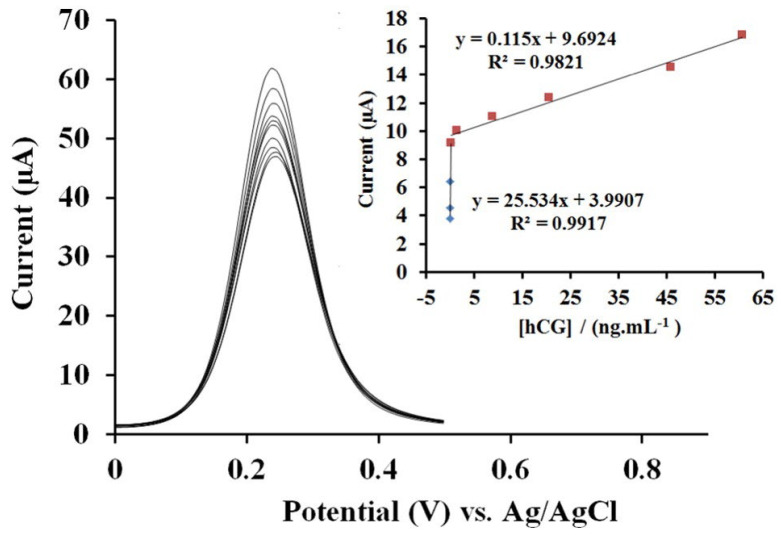
DPV curves of the proposed immunosensor after incubating with different hCG solutions with concentrations of 0, 0.001, 0.01, 0.1, 0.2, 1.48, 8.88, 20.6, 45.8, and 60.7 ng/mL (from top to bottom) in [Fe(CN)_6_]^3−^/^4−^ as electrolyte solution: Inset: calibration plots of the reduction current versus concentration of hCG (reproduced with permission from ref. [74]).

**Figure 4 micromachines-16-01144-f004:**
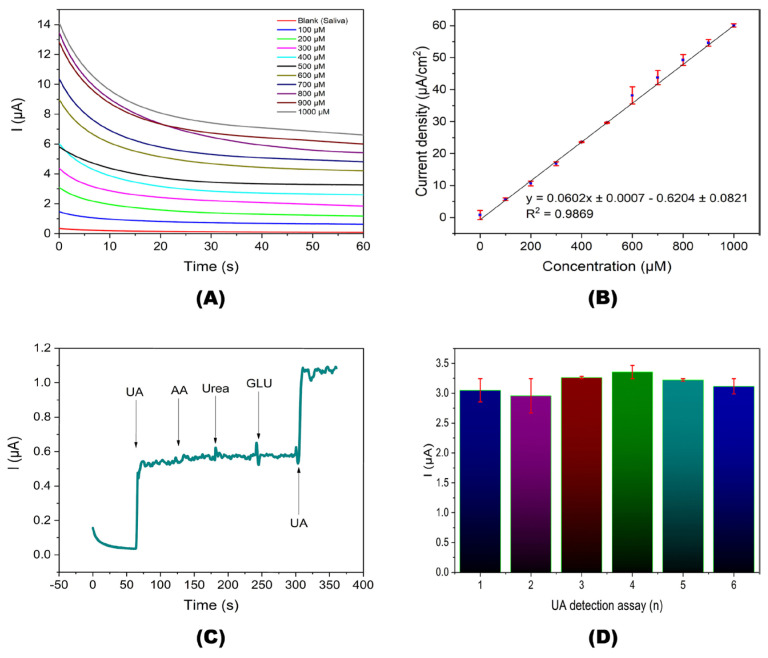
Amperometric detection of UA. (**A**) Amperometric responses at different UA concentrations. (**B**) Lineal fit of the current density depending on UA concentration. (**C**) Continuous selectivity assay in an artificial saliva sample with 20 μM of UA and interfering analytes. (**D**) Repeatability of the amperometric detection at UA concentration of 500 μM (reproduced with permission from ref. [77]).

**Figure 5 micromachines-16-01144-f005:**
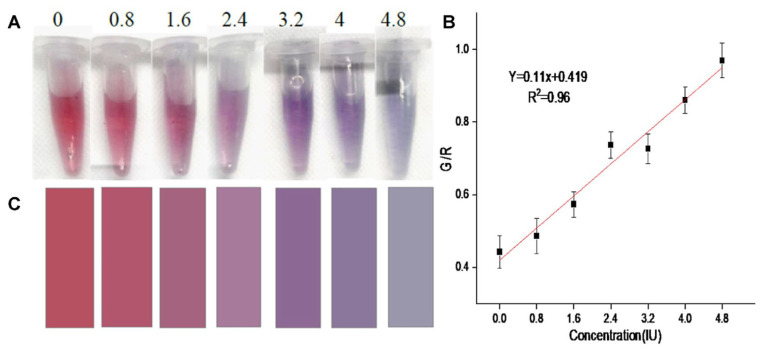
(**A**) Image of CA@AuNPs combined with oxytocin at different concentrations (0.4–4.8 IU). (**B**) Linear graph of G/R ratio and oxytocin concentrations in the range of 0.4–4.8 IU. Error bars represent the standard deviation of three measurements. (**C**) Colorimetric card with mean RGB values extracted from images of added oxytocin in CA@AuNP solution (reproduced with permission from ref. [90]).

**Figure 6 micromachines-16-01144-f006:**
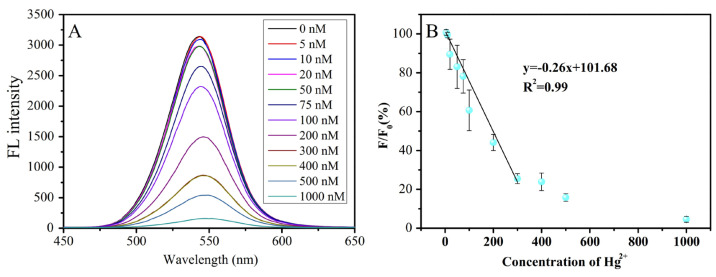
Fluorescence emission spectra of CdTe QDs in the presence of Hg^2+^ at different concentrations in BR buffer at pH 7.54 (**A**), the relationship between F/F_0_ and the concentration of Hg ions (**B**). F and F0 are the FL intensities of QDs at 564 nm in the presence and absence of Hg^2+^ (reproduced with permission from ref. [99]).

**Figure 7 micromachines-16-01144-f007:**
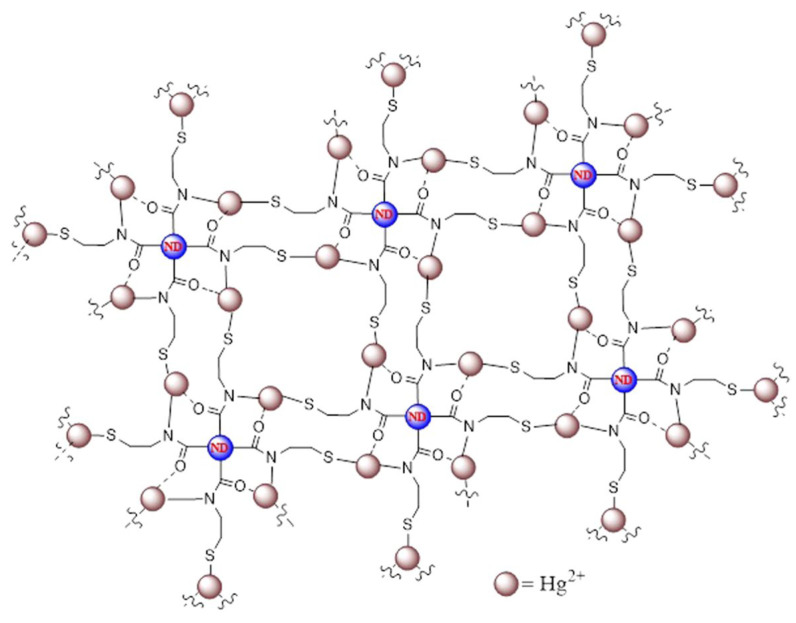
Schematic representation of Hg^2+^-induced agglomeration of CA@NDs (reproduced with permission from ref. [102]).

**Figure 8 micromachines-16-01144-f008:**
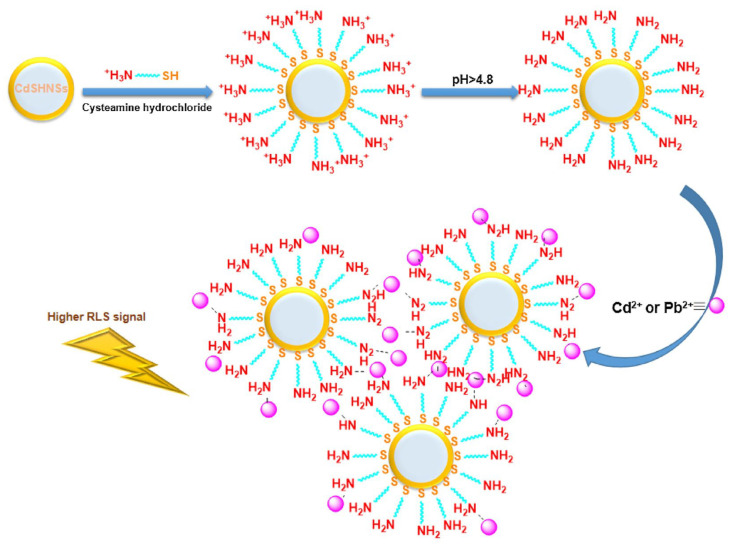
Proposed mechanism for the enhancement of CA@CdSHNSs RLS signal in the presence of Cd^2+^ and Pb^2+^ ions (reproduced with permission from ref. [112]).

**Figure 9 micromachines-16-01144-f009:**
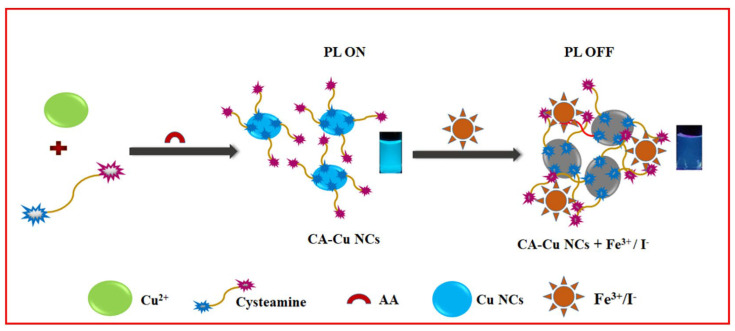
Schematic representation of CA@CuNC synthesis and its sensor responses to Fe^3+^ and I^−^ ions (reproduced with permission from ref. [120]).

**Figure 10 micromachines-16-01144-f010:**
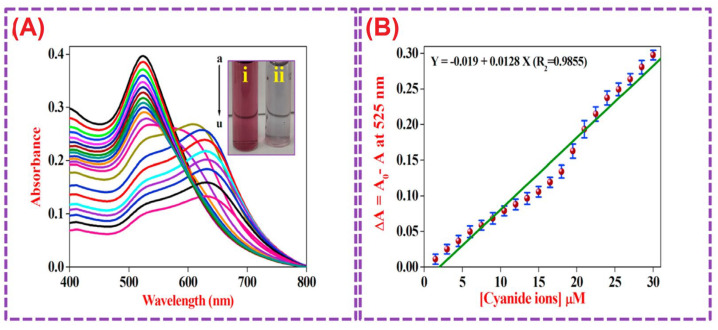
(**A**) UV–vis absorbance spectral data for CA@AuNPs with increasing concentrations of CN^−^ ranging from 1.5 to 30 µM (each addition of 1.5 µM); the inset shows CA@AuNPs and CA@AuNPs in the presence of CN^−^. (**B**) Calibration plot for SPR-based nanosensors. The standard deviations over five repeated measurements at each concentration are shown by the error bars. (Reproduced with permission from ref. [124].)

**Figure 11 micromachines-16-01144-f011:**
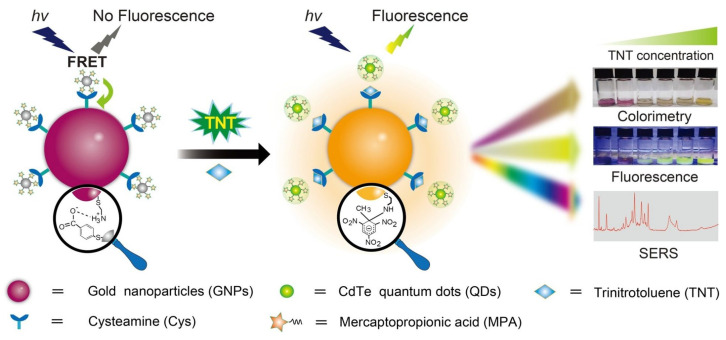
Schematic illustration of a tri-mode sensing platform for TNT analysis enabled by GNPs@QD assemblies (reproduced with permission from ref. [130]).

**Figure 12 micromachines-16-01144-f012:**
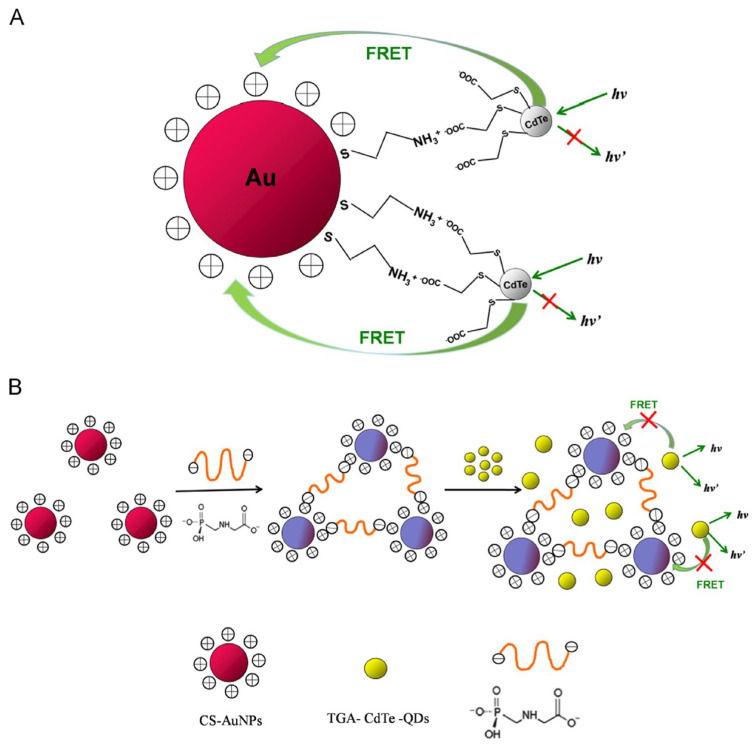
Schematic illustration of (**A**) FRET between TGA-CdTe-QDs and CA@AuNPs, and (**B**) glyphosate-induced attenuation of FRET and fluorescence recovery of quenched QDs (reproduced with permission from ref. [136]).

## Data Availability

Not Applicable.

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
