# Peer review of "A Review on Biomedical, Biomolecular, and Environmental Monitoring Applications of Cysteamine Functionalized Nanomaterials"

_micromachines, 2025, doi:10.3390/mi16101144_

Round 1
Reviewer 1 Report
Comments and Suggestions for Authors
The manuscript entitled “A Review on Biomedical, Biomolecular, and Environmental Monitoring Applications of Cysteamine Functionalized Nanomaterials” provides a comprehensive and timely overview of recent advances in the use of cysteamine-functionalized nanomaterials for sensing applications. The topic is of significant interest to the readers of Micromachines, and the review is generally well-structured and informative. The author successfully summarizes a broad range of biomedical, biomolecular, and environmental applications and highlights both opportunities and limitations. Overall, the manuscript is of good quality and could be accepted after minor revisions. My detailed comments are as follows:
- The manuscript would benefit from careful language polishing to improve readability, as some sentences are lengthy or repetitive. For example, simplifying descriptions of detection ranges and limits of detection (LOD) would improve clarity.
- While the review cites many applications, in some sections the specific mechanistic role of cysteamine is not clearly explained. The author is encouraged to more explicitly highlight how cysteamine contributes to stability, binding, or selectivity in each example.
- Please check that all references are formatted consistently according to journal style (some entries in the text appear with minor inconsistencies in numbering or spacing).
- The conclusion could be strengthened by providing clearer forward-looking perspectives, such as potential commercialization pathways, integration with portable devices, or strategies to overcome stability/cost challenges.
Author Response
The manuscript entitled “A Review on Biomedical, Biomolecular, and Environmental Monitoring Applications of Cysteamine Functionalized Nanomaterials” provides a comprehensive and timely overview of recent advances in the use of cysteamine-functionalized nanomaterials for sensing applications. The topic is of significant interest to the readers of Micromachines, and the review is generally well-structured and informative. The author successfully summarizes a broad range of biomedical, biomolecular, and environmental applications and highlights both opportunities and limitations. Overall, the manuscript is of good quality and could be accepted after minor revisions. My detailed comments are as follows:
I am very grateful to the reviewer for providing the appreciation and opportunity to clarify our results, which improves the standard of this article.
- The manuscript would benefit from careful language polishing to improve readability, as some sentences are lengthy or repetitive. For example, simplifying descriptions of detection ranges and limits of detection (LOD) would improve clarity.
Author Response”
Following the Reviewer’s guidance, the language of this manuscript has been carefully refined. The detection ranges and LODs are essential to highlight the key contributions of each literature. Therefore, this information has been included for the reader’s benefit. Altering it could impact the metrics and readability of this review. Consequently, I believe there is no need to change this information at this stage, as it is necessary for clarification.
- While the review cites many applications, in some sections, the specific mechanistic role of cysteamine is not clearly explained. The author is encouraged to more explicitly highlight how cysteamine contributes to stability, binding, or selectivity in each example.
Author Response”
I appreciate the reviewer's observation regarding the limited exploration of mechanisms in certain reports. It's important to note that the original reports do not provide details on these mechanisms, as highlighted in sections 2.1, 3.6, and 4.2. Without the authors' consent, I am unable to speculate on precise mechanisms. However, I have ensured that the mechanisms are thoroughly clarified in this review wherever it is necessary. I trust the reviewer will consider this explanation. Thank you for your understanding.
- Please check that all references are formatted consistently according to journal style (some entries in the text appear with minor inconsistencies in numbering or spacing).
Author Response”
Thanks for your valuable advice. The format style of all the references has been checked carefully with the guidelines of the manuscript.
- The conclusion could be strengthened by providing clearer forward-looking perspectives, such as potential commercialization pathways, integration with portable devices, or strategies to overcome stability/cost challenges.
Author Response”
As per the Reviewer’s suggestion, the conclusion section is boosted with the following points to address the issue.
“(F) Use of contaminant free injectable and portable microfluidic chips and devices could be fabricated with cysteamine-functionalized nanomaterials to overcome stability/cost challenges [161]; (G) Combinations of Artificial Intelligence (AI) [162], machine learning [163], and density functional theory (DFT) studies may boost the monitoring features of cysteamine-functionalized nanomaterials to deliver state-of-the-art commercial products for day-to-day life.”
Reviewer 2 Report
Comments and Suggestions for Authors
This review “A Review on Biomedical, Biomolecular, and Environmental Monitoring Applications of Cysteamine Functionalized Nanomaterials” describes the use of cysteamine as a binding agent for preparing nanomaterials used in analytical chemistry to detect a wide range of analytes. The author focus on applications in biomedical, biomolecular, and environmental fields. The manuscript also outlines the advantages, disadvantages, and future prospects of these analytical systems. While the research is of high quality, some areas require improvement, as detailed below:
1. To improve clarity, please include a summary table that lists the key characteristics of the nanomaterials discussed (e.g., target analyte, limit of detection (LOD), linear range).
2. The abstract states that cysteamine functionalization significantly improves nanomaterial stability. However, the limitations section notes that insufficient stability remains a barrier to commercialization. This apparent contradiction should be addressed to better frame the review's discussion.
Author Response
This review “A Review on Biomedical, Biomolecular, and Environmental Monitoring Applications of Cysteamine Functionalized Nanomaterials” describes the use of cysteamine as a binding agent for preparing nanomaterials used in analytical chemistry to detect a wide range of analytes. The author focuses on applications in biomedical, biomolecular, and environmental fields. The manuscript also outlines the advantages, disadvantages, and future prospects of these analytical systems. While the research is of high quality, some areas require improvement, as detailed below:
I am grateful to the reviewer for providing valuable comments, which have enhanced the quality of this manuscript.
- To improve clarity, please include a summary table that lists the key characteristics of the nanomaterials discussed (e.g., target analyte, limit of detection (LOD), linear range).
Author Response”
Thanks for your valuable suggestion. This is not possible at this stage with respect to the timeline provided. Further, inclusion of such a table might lengthen the review and affect the readability and metrics. Since the sections are divided into subsections, I feel this is not necessary at this stage
- The abstract states that cysteamine functionalization significantly improves nanomaterial stability. However, the limitations section notes that insufficient stability remains a barrier to commercialization. This apparent contradiction should be addressed to better frame the review's discussion.
Author Response”
Thanks for your valuable suggestion. In limitations, it has been stated that the stability may be affected in harsh environmental conditions. In abstract, there are no conditions, such as harsh environmental conditions are not mentioned. There is no contradiction.
Reviewer 3 Report
Comments and Suggestions for Authors
Reviewer report
Reviewer report on Manuscript Draft ‘A Review on Biomedical, Biomolecular, and Environmental Monitoring Applications of Cysteamine Functionalized Nanomaterials ’
This review emphasizes the role of cysteamine in producing stable nanomaterials and detecting specific biomedical, biomolecular, and ecological analytes. It also covers general protocols for functionalizing with cysteamine, the mechanistic basis of analyte detection, and their advantages, limitations, and prospects.
This manuscript is well-written, well-illustrated, and interestingly addressed. The manuscript is in the scope of the journal; therefore, eventually, it can be published after some improvements and corrections:
Chapter ‘2. Biomedical and biomolecular monitoring applications’ first paragraph could be extended by references (Electrochemical Biosensor for the Evaluation of Monoclonal Antibodies Targeting the N Protein of SARS-CoV-2 virus. Science of the Total Environment, 2024, 924, 171042.) where self-assembled monolayers are applied for the immobilization of SARS-CoV-2 virus proteins. Advantages and disadvantages of cysteamine-based protein immobilization could be compared with those where self-assembled monolayers are applied for the immobilization of proteins.
Chapter 4.1. Limitations should be advanced by limitations related to the stability of cysteamine-based structures in comparison to those that are based on longer -CH2- chain containing thiols, which form more stable self-assembled monolayers.
Author Response
This review emphasizes the role of cysteamine in producing stable nanomaterials and detecting specific biomedical, biomolecular, and ecological analytes. It also covers general protocols for functionalizing with cysteamine, the mechanistic basis of analyte detection, and their advantages, limitations, and prospects. This manuscript is well-written, well-illustrated, and interestingly addressed. The manuscript is in the scope of the journal; therefore, eventually, it can be published after some improvements and corrections:
I am grateful to the reviewer for providing valuable comments, which have enhanced the quality of this manuscript.
Chapter ‘2. Biomedical and biomolecular monitoring applications’ first paragraph could be extended by references (Electrochemical Biosensor for the Evaluation of Monoclonal Antibodies Targeting the N Protein of SARS-CoV-2 virus. Science of the Total Environment, 2024, 924, 171042.) where self-assembled monolayers are applied for the immobilization of SARS-CoV-2 virus proteins. Advantages and disadvantages of cysteamine-based protein immobilization could be compared with those where self-assembled monolayers are applied for the immobilization of proteins.
Author Response”
Thanks for your valuable suggestion. The aim and objective of this review is to deliver valuable information on the use of cysteamine-functionalized nanomaterials in biomedical, biomolecular, and environmental screening, and not to compare with other tactics. Since other tactic also has certain advantages. At this stage, it is not possible to include the suggested reference in Chapter 2 and provide a comparative account, which would involve comparing each report with other available reports in this research. Further, the suggested report does not contain any cysteamine-related information, which limits its inclusion in Chapter 2. However, I cited that reference in Section 4.1, "Advantages," with the following text.
“(3) Use of CA@AuNPs for optical reflectometry and colorimetry-based detection of SARS-CoV-2 (COVID-19) [51] has similar advantages to an electrochemical sensor driven by SAM facilitated immobilization of SARS-CoV-2 virus proteins [150].”
Chapter 4.1. Limitations should be advanced by limitations related to the stability of cysteamine-based structures in comparison to those that are based on longer -CH2- chain containing thiols, which form more stable self-assembled monolayers.
Author Response”
Thanks for your valuable suggestion. Regarding the stability of longer -CH2- chain containing thiols in self-assembled monolayer compared to cysteamine has been mentioned in section 4.2 Limitations as noted below.
“(3) The SAM-based sensor performance depends on its stability, which may be greater in longer -CH2- chain containing thiols than cysteamine [160]. This could limit the design of such sensors.”